# Surface Bacterioplankton Community Structure Crossing the Antarctic Circumpolar Current Fronts

**DOI:** 10.3390/microorganisms11030702

**Published:** 2023-03-09

**Authors:** Angelina Cordone, Matteo Selci, Bernardo Barosa, Alessia Bastianoni, Deborah Bastoni, Francesco Bolinesi, Rosaria Capuozzo, Martina Cascone, Monica Correggia, Davide Corso, Luciano Di Iorio, Cristina Misic, Francesco Montemagno, Annarita Ricciardelli, Maria Saggiomo, Luca Tonietti, Olga Mangoni, Donato Giovannelli

**Affiliations:** 1Department of Biology, University of Naples Federico II, 80126 Naples, Italy; 2Dipartimento di Scienze della Terra, Dell’Ambiente e della Vita, Universitá di Genova, 16132 Genova, Italy; 3Stazione Zoologica Anton Dohrn, 80121 Naples, Italy; 4Department of Science and Technology, University of Naples Parthenope, 80143 Naples, Italy; 5Consorzio Nazionale Interuniversitario delle Scienze del Mare (CoNISMa), 00196 Rome, Italy; 6Institute of Marine Biological Resources and Biotechnologies, National Research Council, 60125 Ancona, Italy; 7Earth-Life Science Institute, Tokyo Institute for Technology, Tokyo 152-8552, Japan; 8Department of Marine and Coastal Science, Rutgers University, New Brunswick, NJ 08901, USA; 9Marine Chemistry and Geology Department, Woods Hole Oceanographic Institution, Woods Hole, MA 02540, USA

**Keywords:** bacterioplankton diversity, Southern Ocean, 16S rRNA sequencing, primary productivity

## Abstract

The Antarctic Circumpolar Current (ACC) is the major current in the Southern Ocean, isolating the warm stratified subtropical waters from the more homogeneous cold polar waters. The ACC flows from west to east around Antarctica and generates an overturning circulation by fostering deep-cold water upwelling and the formation of new water masses, thus affecting the Earth’s heat balance and the global distribution of carbon. The ACC is characterized by several water mass boundaries or fronts, known as the Subtropical Front (STF), Subantarctic Front (SAF), Polar Front (PF), and South Antarctic Circumpolar Current Front (SACCF), identified by typical physical and chemical properties. While the physical characteristics of these fronts have been characterized, there is still poor information regarding the microbial diversity of this area. Here we present the surface water bacterioplankton community structure based on 16S rRNA sequencing from 13 stations sampled in 2017 between New Zealand to the Ross Sea crossing the ACC Fronts. Our results show a distinct succession in the dominant bacterial phylotypes present in the different water masses and suggest a strong role of sea surface temperatures and the availability of Carbon and Nitrogen in controlling community composition. This work represents an important baseline for future studies on the response of Southern Ocean epipelagic microbial communities to climate change.

## 1. Introduction

Oceans cover more than 70% of Earth’s surface, driving almost half of the global net primary production [1,2,3]. The Southern Ocean, in particular, is thought to contribute to nearly 15% of the oceanic primary production [4], and it is generally defined as the global ocean south of 60° S surrounding the Antarctic continent [5]. Due to the upwelling of nutrient-rich Circumpolar Deep Waters (CDW), as well as the presence of water masses with different physicochemical properties, the Southern Ocean plays an important role in the ocean ecosystem’s functioning and composition [5,6].

The Southern Ocean is home to the world’s strongest and deepest currents that control thermohaline circulation across the entire world’s oceans. It connects the three main ocean basins (Atlantic, Pacific and Indian) and creates a global circulation system that is largely driven by the Antarctic Circumpolar Current (ACC). The ACC flows from west to east around Antarctica, and it is considered the major water exchange driver between the world’s oceans, being the only current that flows entirely around the globe. The ACC consists of a number of full-depth fronts, defined by changes in water properties that occur in a short distance, characterized by high velocities and separated by relatively quiescent zones [7]. These frontal regions also represent a strong exchange between waters, enabling the upwelling of deep nutrient-rich waters and the downwelling of less enriched surface waters. Hence, ACC fronts are considered to play an important role in the global distribution of nutrients (i.e., nitrate) due to the vertical exchange within fronts and the “mixing barrier” effect across fronts [5].

Traditionally, the Southern Ocean has been divided into three main fronts (from north to south): the Subantarctic Front (SAF), the Polar Front (PF), and the Southern ACC Front (sACCF). The northern and southern limits of the ACC are marked by the Subtropical Front (STF) and the Southern Boundary Front (SBDY) [5,8,9], respectively. Both fronts have very distinct dynamics with respect to the main ACC fronts, leading some authors to not consider them as part of the main ACC [7,10]. The ACC fronts are at their narrowest meridional constriction at the Drake Passage and diverge as the ACC flows downstream into the Scotia Sea. The island of South Georgia lies on the northeastern side of the Scotia Sea. It is south of the PF but is strongly influenced by the SACCF that loops anti-cyclonically around the island’s shelf from the south before retroflecting to the east. Before reaching South Georgia, the SACCF runs through the southern half of the Scotia Sea, and although the island itself lies in the polar open ocean zone, the waters that lie off the north coast may have been seasonally influenced by the presence of ice cover.

Despite the persisting high nutrient and low chlorophyll (HNLC) conditions [11], primary productivity within the ACC is highly variable. Maximum phytoplankton growth rates are relatively low due to a combination of low temperatures, micronutrient availability, and high wind stress leading to a strong mixing over the region. However, the frontal zones’ primary production generally results elevated compared to surrounding oceanic areas [12,13,14] and can be spatially extensive. Different studies show that the enhanced production in these regions is correlated to increased supplies of iron derived from shelf sediments, which are available within frontal jets [12,15]. In contrast, oceanic areas to the north and south of the fronts are iron-limited, and the low chlorophyll levels have a negative impact on export production [16,17].

The microbial diversity of such environments is shaped by both geography and environmental variables [18]. Physical barriers in marine environments are less evident compared to those in terrestrial settings (such as mountains, oceans, and islands) but are still remarkable [19]. The Antarctic Polar front (APF) [20,21], for instance, can create an open ocean dispersal barrier due to intense currents and a 3–4 °C horizontal thermocline [22,23]. The presence of fronts, such as the APF, has been shown to influence the genetic flow for larger eukaryotes such as Echinoderms, Annelids, and Chordates (e.g., brittle stars, worms, and toothfish) [22,24,25]. Prokaryote communities also appear to be separately shaped by ocean fronts [6,23,26], and the spatial distribution of phytoplankton groups seems to be highly correlated to ocean surface thermal gradients across the ACC [27].

Marine microorganisms, especially Bacteria and Archaea, are fundamental for the functioning of the world’s oceans as they are the main players in the biogeochemical cycling of the elements. Their diversity and abundance have been extensively studied during the last decades thanks to different oceanographic campaigns that allowed for the collection of several seawater and sediment samples. For instance, the Tara Ocean Expedition, started in 2009, has allowed for more than 30,000 samples to be collected in more than 200 stations distributed throughout the world’s oceans [28].

16S rRNA sequencing and shotgun metagenomic analysis from these samples revealed that the bacterial community of the world’s oceans is dominated by Proteobacteria, like clades SAR11 and SAR86, both in terms of relative abundance and taxonomic richness, followed by Cyanobacteria, Deferribacteres, and Thaumarcheaota [29]. Despite the great effort in analyzing the ocean’s microbiome, the microbial diversity of the Southern Ocean, especially related to the water masses crossing this region, is still poorly characterized.

The rapid warming of the Southern Ocean could influence species turnover in the microbial community with widespread consequences for the global climate [6]. In this context, our study aims to characterize the bacterioplankton community of surface waters crossing the Antarctic Circumpolar Current Fronts from New Zealand to the Ross Sea and to investigate the effect of physicochemical changes of these fronts on the bacterial community structure.

## 2. Materials and Methods

### 2.1. Sampling Procedure and Study Site

Sampling activities were carried out on the R/V Italica during the austral summer of 2017 in the framework of a plankton biodiversity and functioning of the Ross Sea ecosystems in a changing Southern Ocean (P-ROSE) project funded by the Ministry of Education, University and Research (MIUR). Surface water samples (approximate depth of 5 m) were collected along the transect New Zealand–Ross Sea every 8 h using a plastic tube pumping water on board (Figure 1a,b). Seawater aliquots of 500 mL were filtered (Whatman 0.22 μm) and kept frozen (between −20 and −80 °C) on board. Of the 24 samples recovered, high-quality 16S rRNA libraries were obtained only for 13 stations and were used for downstream analysis.

Further samples were collected for the biochemical measurements of particulate organic matter. Water (at least 1000 mL in duplicate for each analysis) was filtered through Whatman GFF filters (glass fiber, nominal porosity 0.45 µm), and stored at −20 °C until analysis.

### 2.2. Environmental Parameters

For each sampling station, data for Surface Primary Productivity (SPP) were obtained from the Ocean Productivity Database [30], based on the generalized vertical production model developed by [31]. Sea Surface Temperature (SST), Sea Surface Temperature Anomalies (SSTA), and Wind Speed (WS) data were obtained from the National Oceanic and Atmospheric Administration database [32], while data for the Current Speed (CS) were obtained from the NASA-funded research project, Ocean Surface Current Analysis Real-Time (OSCAR) [33]. Sea Surface Salinity (SSS) data were obtained from the Physical Oceanography Distributed Active Archive Center [34].

### 2.3. Community DNA Extraction

DNA extraction was carried out according to Giovannelli et al. (2016) with minor modifications (extra elution steps). Briefly, each filter was soaked in a saline lysing solution (100 mM NaCl (pH 8.0), 20 mM EDTA, 50 mM Tris-HCl (pH 8.0)) to which 50 µL of lysozyme (100 mg/mL) was added. The resulting solution was incubated at 37 °C for 1 h and treated with 50 µL of Proteinase K (20 mg/mL), followed by an incubation step at 37 °C for 1 h with shaking every 15 min. Subsequently, 1 mL of 20% sodium dodecyl sulfate (SDS) was added to the resulting solution, and the sample was incubated at 65 °C for 1 h. Afterward, samples were centrifuged at 13.000 rpm × 15 min. The supernatant was transferred in a sterile 2 mL Eppendorf tube. Nucleic acids were separated from the other cellular components by performing two consecutive liquid:liquid extractions with phenol:chloroform:isoamylic alcohol (25:24:1), each followed by a centrifugation step of 15 min at room temperature. After obtaining a phase separation, DNA was precipitated by adding 0.7 volumes of isopropanol (100%) and 0.1 volumes of sodium acetate and incubated overnight at room temperature. Finally, samples were centrifuged at 13.000 rpm × 30 min. The resulting pellet was washed with ethanol and re-precipitated by centrifuging at 13.000 rpm × 5 min. After drying the pellet at room temperature, the obtained DNA was resuspended with 50 µL of Tris-HCl (50 mM, pH 8.0). The effectiveness of the extraction was evaluated after performing runs on the electrophoresis cell, using 1% agarose gel stained with ethidium bromide, and visualizing the gel at the UV transilluminator [35,36,37]. The DNA integrity was checked spectrophotometrically and by PCR amplification. Briefly, a 500 bp fragment of the 16S rRNA gene was PCR amplified with MyTaq DNA Polymerase (Bioline) by using total DNA from each sample as a template and the oligo Ribo-For (5′-AGTTTGATCCTGGCTCAG-3′) and Ribo-rev (5′-ACCTACGTATTACCGCGGC-3′) as primers. PCR conditions were: 5 min at 95 °C, followed by 30 cycles of 95 °C for 30 s, 50 °C for 30 s, 72 °C for 30 s, concluding with an extension at 72 °C for 5 min. The PCR products were analyzed as previously reported [38,39].

### 2.4. 16S rRNA Gene Sequencing

The obtained DNA was sequenced at the Integrated Microbiome Resource (IMR, [40]) using primers targeting the V4-V5 of the 16S rRNA (515FB = GTGYCAGCMGCCGCGGTAA 926R = CCGYCAATTYMTTTRAGTTT), using Illumina MiSeq technology.

### 2.5. Biochemical Measurements

Particulate Organic Carbon (POC) and Particulate Nitrogen (PN) were analyzed as previously reported [41]. Briefly, filters were exposed to fumes of HCl for 4 h to remove inorganic carbon [41]. After drying (60 °C), filters were placed into tin capsules and analyzed using a Carlo Erba Mod. 1110 CHN Elemental Analyzer (dynamic flash combustion). Cyclohexanone 2–4-dinitrophenyl hydrazone (purchased from Sigma-Aldrich, Steinheim, Germany) was used as a reference standard. For the determination of fractionated Chlorophyll, a protocol of serial filtration was followed, as reported by Mangoni et al. [42]. Frozen filters were processed in Italy for the determination of Chl a and phaeopigments (phaeo) content, using a solution of 90% acetone according to [43,44], with a spectrofluorometer (Shimadzu, Mod. RF–6000; Shimadzu Corporation-Japan) checked daily with a Chl a standard solution (Sigma-Aldrich). Particulate carbohydrate and protein concentrations were determined following [45,46], respectively. The method for carbohydrate determination is based on the reaction of the carbohydrates to phenol (5% in water solution) in an acid medium (sulfuric acid). For protein determination, copper sulfate pentahydrate was added to the samples, leading to the bond of Cu to amino acids. The Folin–Ciocalteu′s phenol reagent completes the reaction with the development of a blue coloration. A Jasco V-530 spectrophotometer was calibrated with D-glucose solutions for carbohydrates (absorbance 490 nm) and with bovine serum albumin solutions for proteins (absorbance 650 nm). All materials were purchased from Sigma-Aldrich (Steinheim, Germany).

### 2.6. Bioinformatics and Statistical Analyses

Raw reads received from the sequencing center were processed using the DADA2 package [47]. Primers and adapters were trimmed, followed by a quality profile step, where only sequences with a call quality for each base between 20 and 40, were kept for further analysis. Amplicon sequence variants (ASVs) were estimated through the error profile and assigned taxonomy with the SILVA database (release 138) [48]. The resulting taxonomic assignments, in combination with variant abundance tables, were used to create a phyloseq object with the phyloseq package [49], as previously described in [38]. Subsequently, sequences related to Chloroplasts, Mitochondria, and Eukaryotes, as well as groups related to human pathogens and common DNA extraction contaminants [50], were removed from the dataset. The remaining reads represented 53.4% of the original raw reads with 535 ASVs. Alpha diversity was investigated using the Observed, Shannon, and Chao1 diversity indexes, while beta diversity was investigated using the Jaccard diversity index. Additionally, correlations between measured environmental variables and the resulting ordination were achieved using environmental fitting (env_fit function in vegan). The statistical analysis conducted, as well as data processing and visualization, was carried out in R software, version 4.1.2 [51], using the vegan [52] and ggplot2 [53] packages. The sequences analyzed in the present study are publicly available through the European Nucleotide Archive (ENA) with a bioproject accession number PRJEB45048. A complete R script describing the analysis can be found in the GitHub repository https://github.com/giovannellilab/Cordone_et_al_Southern_Ocean_microbial_diversity, accessed on 30 January 2023 and released with Zenodo with https://doi.org/10.5281/zenodo.7584581, accessed on 30 January 2023.

## 3. Results

### 3.1. Environmental and Biogeochemical Parameters

The coordinates of the sampled points and the respective environmental parameters, such as sea surface temperature (SST), sea surface temperature anomaly (SSTA), sea surface salinity (SSS), current speed (CS), and wind speed (WS), are listed in Table 1. There is a steep reduction in sea surface temperature when moving from New Zealand (7.9 °C) to the Antarctic region (−1.8 °C), showing a clear trend with the changes in latitude (Appendix A). As for the sea surface salinity, it decreases moving towards the Antarctic region and then increases again once within the Ross Sea area (Appendix A).

The biogeochemical parameters determined for all sequenced stations are shown in Table 2. Among the biogeochemical parameters analyzed, only the C fraction of dissolved organic matter (C-DOM), total carbohydrates (cho_tot) and particulate nitrogen (PN) showed significant correlations with the geographical location (latitude) of the sequenced stations. As expected, Particulate Organic Carbon (POC) was significantly correlated to PN and total protein content (pr_tot), while pr_tot was also significantly correlated to surface primary productivity (spp), and pn was also correlated to dissolved organic carbon (C-DOM) (Appendix A).

### 3.2. Diversity of Bacterial Communities

Bacterial diversity along the transect New Zealand–Ross Sea was evaluated using the 16S rRNA gene sequence as reported in the materials and methods section. A total of 130,906 reads were obtained after quality check and data filtering and were used to identify 535 unique ASVs. The number of reads, ASVs, phyla, classes, orders, families, and genera per station has been reported in Appendix A.

The rarefaction curves for the total reads–ASVs relationship were created to calculate the coverage of retrieved ASVs over the predicted ASVs obtained using the value of the Chao1 alpha diversity index (Appendix A). The coverages for all the sampled stations (Appendix A) show that the sequencing depth reached in our study is adequate to analyze and characterize the bacterioplankton community structure crossing the ACCF, ranging from 83% and 96%. The alpha diversity evidenced a decreasing trend from the stations belonging to STF until SACCF, which sharply increased towards the southernmost part of the transect (Appendix A).

As shown in Figure 2, in the station belonging to STF (stations 5, 6, 7, and 10), we mainly found Proteobacteria (recently reclassified as Pseudomonadota), Bacteroidota, and Cyanobacteria with an average of 44.78%, 40.94%, and 9.64% of reads assigned, respectively. Station 12, present on the SAF, had a similar distribution, with Proteobacteria (67.43%) and Bacteroidota (27.36%) being the two most abundant phyla, followed by Myxococcota (0.35%). Reads from the only station influenced by the PF (station 15) were also assigned to Proteobacteria (58.85%), Bacteroidota (38.92%), and Cyanobacteria (0.57%). In the case of station 16, associated with the SACCF, 57.16% of the reads were assigned to Bacteroidota, while 41.10% corresponded to Proteobacteria. In stations 17, 18, 19, and 20, all influenced by the SBDY, we mainly found Proteobacteria (50.90%) and Bacteroidota (47.84%), with Verrucomicrobiota having assigned just 0.12% of the reads. Finally, the two stations (23 and 24) belonging to the RSCS were also characterized by a predominance of Bacteroidota (49.57%) and Proteobacteria (46.69%), with nearly 0.1% of the reads (0.06%) assigned to the SAR324 clade (Marine group B).

Within the Bacteroidota, the Bacteroidia class was the most abundant among the stations influenced by the STF, SACCF, and RSCS, with average reads assigned of 36.91%, 57.16%, and 55.64%, respectively, while the stations influenced by the SAF, SPF, and SBDY, had an average of reads assigned equal to 27.36%, 38.92%, and 33.80%, respectively. At the order level, Flavobacteriales represent the most abundant order, similarly distributed in all stations (Figure 3A), with the exception of station 12 (influenced by the SAF front), with an average of reads assigned of 44.43%, 38.66%, 55.06%, 46.59%, 45.91%, and 24.36% to the stations influenced by STF, SPF, SACCF, SBDY, RSCS, and SAF, respectively.

A different distribution is observed for the Cytophagales and Chitinophagales orders, with the Cytophagales mainly detected in station 12 (2.86%) localized in the Sub-Antarctic front, and the Chitinophagales mainly present in the southern area of the transect—on average 3.31% for stations within the RSCS. Among the Flavobacteriales, different genera can be distinguished. The stations in the south are dominated by sequences assigned to the genera *Polaribacter* (40.78% in station 16 and 26% in the RSCS stations) and *Aurantivirga* (5.78% in the RSCS stations), while the stations in the north, influenced by STF, SAF, and SPF, are dominated by sequences assigned to the genera *Formosa*, with an average read assigned equal to 4%, 4.20%, and 2.12%, respectively. The NS2b marine group is present in all the stations analyzed (Appendix A).

Within the Proteobacteria, most sequences are affiliated with the classes Alphaproteobacteria and Gammaproteobacteria. Among Alphaproteobacteria, SAR11 clade (STF = 16.25%, SAF = 27.95%, PF = 24.27%, SACCF = 5.39%, SBDY = 14.56%, RSCS = 12.26%) and Rhodobacterales (STF = 6.22%, SAF = 18.54%, PF = 14.42%, SACCF = 13.78%, SBDY = 10.05%, RSCS = 11.8%) represent the most abundant orders in all the stations, whereas Puniceispirillales are mainly present in the northern part of the transect, with a maximum of 5.68% in station 12 (SAF front) and a minimum of 1.55% in the PF (Figure 3B). Gammaproteobacteria are mainly represented by the orders SAR86 clade (STF = 4.63%, SAF = 2.99%, PF = 3.06%, SACCF = 0.20%, SBDY = 1.99%, RSCS = 1.25%) more abundant in the northernmost part of the transect, and Cellvibionales (STF = 0.85%, SAF = 5.09%, PF = 6.88%, SACCF = 6.51%, SBDY = 4.93%, RSCS = 8.59%), which instead are mainly distributed in the southernmost part. The order Thiomicrorpirales is detected in all the stations, with average reads assigned varying from 0.51% in station 15 (within PF) to 6.70% for stations 17, 18, 19, and 20 (SBDY). Stations within the STF had an average of 3.96%, while station 12 (SAF) had 1.33%, station 16 (SACCF) 1.01%, and stations 23 and 24 (RSCS) had an average of 4.18%. The orders Oceanospirillales and Alteromonadales are mainly detected in the southernmost stations (Figure 3C), with the highest percentage of average reads assigned to Oceanospirillales (8.28%) to the station within the SACCF (station 16) and the lowest (0.39%) to the stations belonging to the STF (stations 5, 6, 7 and 10); while Alteromonadales where mostly assigned to the reads from the SACCF and the SBDY stations (4.87% and 4.03%, respectively) (Figure 3C) with stations from the STF, SAF, PF, and RSCS having lower averages of assigned reads (0.09%, 0.12%, 1.64%, and 1.07%, respectively).

As mentioned before, sequences belonging to the phylum Cyanobacteria were detected mainly in the four northern stations of the transect (9.6% relative abundance), all influenced by the STF. In particular, within this phylum, most sequences are affiliated with the Synechococcales order (Figure 4).

To explain the bacterial community structure across the different fronts, we performed a principal coordinate analysis (PCoA) based on the relative abundance of the assigned ASVs to each station and generated using a weighted Jaccard similarity distance. The first two axes of the PCoA were able to explain about 63% of the variance. As shown in Figure 5, there is a “U” pattern along Axis 1, also known as a “horseshoe” effect. This pattern has been associated with species turnover along an environmental gradient [54]. The site distribution follows a latitudinal trend, from north to south, with the only exception of station 16. The ASVs that mainly contribute to the distribution of our samples within the PCoA ordination are shown in Figure 6. Bacteroidota and Proteobacteria are distributed within the whole ordination, together with Cyanobacteria, Verrucomicrobia and Actinobacteriota, while the left part of the ordination is more characterized by Marinimicrobia, Thermoplasmatota, and Bdellovibrionota.

To further investigate the differences between the sampling sites and to discern the influence of the biogeochemical parameters on the community structure, we performed a principal component analysis (PCA). The PCA was able to explain 77.2% of the total variance associated with our community. A bi-plot of the PCA obtained is shown in Figure 7 and confirms the presence of a gradient between the different sites. In particular, samples associated with the STF (northern sampling sites) are influenced by sea surface temperature (sst), latitude (lat), and the chlorophyll-a picofraction (pico_chla). On the other hand, samples associated with the most southern fronts (PF, SBDY, and RSCS) seem to be more influenced by variables such as the carbon fraction of dissolved organic matter (C-DOM) and total carbohydrates (cho_tot). Only the sample from station 24 and associated with the RSCS is more influenced by parameters like particulate nitrogen (PN), total protein content (prt_tot), sea primary productivity (spp), and particulate organic carbon (POC) contents of the seawater.

## 4. Discussion

The Southern Ocean is the least studied oceanic region of the world, mainly as a consequence of its harsh climate and remoteness, making systematic sampling campaigns both difficult and expensive to conduct [5,55]. Water masses characterizing the Southern Ocean have different physicochemical properties, particularly surface temperature and salinity, which allow for their division into fronts [5,8,9]. These differences have led researchers to hypothesize the establishment of a zonation within the distribution of the microbial communities, with fronts playing a key role in structuring the microbial ecosystem. Our results supported this idea as the PCoA analysis shows a horseshoe pattern usually associated with species turnover over environmental gradients. The environmental gradients recovered by the PCA analysis follow the same general trend as the PCoA, suggesting a tight coupling between environmental variations across the transect and species turnover. In fact, stations that fall under the influence of the same front were clustered together and influenced by the same set of environmental variables, with sea surface temperature being among the strongest drivers and having a significant correlation with the PCoA axis 1 (*p* < 0.005).

We also observed a significant change in the phyla present as we moved further south and away from the influence of the SubTropical Front (STF), where warmer waters favor the presence of Cyanobacteria alongside members of the Proteobacteria and Bacteroidota phyla. Proteobacteria are mainly represented by Alphaproteobacteria and Gammaproteobacteria, while within Bacteroidota, the Flavobacteriales order was detected in all the stations with higher abundance. The presence of these phyla is consistent with previously reported results [6,56,57]. Additionally, the stations under the influence of the STF current are the only ones where the Cyanobacteria phylum is present, with an average abundance of 9.6% in stations 5, 6, 7, and 10

Among Alphaproteobacteria, SAR11 is thought to be the dominant bacterial group in Southern Ocean waters regardless of depth and distance from shore; however, it appears to be more abundant in the epipelagic zone [6]. Our results are in agreement with the biogeographic partitioning reported by several authors [6,58], where the abundance of SAR11 is higher near the Sub-Artic Front (SAF) and the Polar Front (PF) and decreases towards the Antarctic zones. According to Wilkins and coauthors [6], this behavior could be a consequence of the oligotrophic nature of SAR11, which gives it a competitive advantage in a High Nutrient–Low Chlorophyll (HNLC) environment compared to the Antarctic zone where phytoplankton blooms increase the concentration of high molecular weight dissolved organic matter and organic particulates. Among Gammaproteobacteria, the SAR86 clade follows the same trend as SAR11, as its presence is generally detected in HNLC waters [59], while Alteromonadales and Oceanospirillales, comprising heterotrophic cold-adapted bacteria, were mainly detected at the stations between the Southern ACC front and the Ross Sea continental shelf, where the concentration of particulate organic carbon was higher. This result is consistent with other reports suggesting that these taxa are major players in the POC remineralization process [60]. Bacteroidota were mainly represented by members of the Flavobacteriales order, which includes members capable of degrading High Molecular Weight (HMW) DOM and whose presence is generally associated with the phytoplankton blooms in such an area [6,61,62]. In particular, within Flavobacteriales, members belonging to *Polaribacter* and *Formosa* genus showed an interesting trend with an increase in the relative abundance of ASVs assigned to *Polaribacter* when approaching the RSCS and a concomitant decrease in the relative abundance of ASVs assigned to the *Formosa* genus. Abell and Bowman [63] hinted at a possible influence of water temperature over the diversity of the Flavobacteria clades in the Southern Ocean, showing that samples from the temperate zone and the Sub-Antarctic zone had relatively lower Flavobacteria diversity than samples from the Polar Front zone and the Antarctic zone. We also observed that there was a higher relative abundance of Flavobacteria in stations where the Alphaproteobacteria clade was at its lowest abundance. According to the literature [63,64,65,66,67], this trend could be a consequence of the differential degradation of organic matter capacity exhibited by both groups of organisms, where Flavobacteria are able to produce proteins used to degrade complex organic substrates while Alphaproteobacteria SAR11 have the capability of synthesizing high-affinity uptake systems that allow the utilization of simpler byproducts of algal polymer degradation (i.e., sugars, acetate, ammonia), as metaproteomic assessments showed [67].

Recent evidence has shown an increase in cyanobacterial communities in response to increasing seawater temperatures [68]. Our results further support this trend since we are detecting Cyanobacteria only in the stations with higher temperatures and under the influence of the STF. Given the increase in Antarctic seawater temperatures [69], we could expect a future species turnover in the microbial communities inhabiting the different Antarctic current fronts, similar to what we found in this study. However, the consequences of this turnover in the overall community structure and trophic levels are still not known.

Taken together, our results show a clear ecological succession of bacterial taxa linked to the oceanic fronts crossing the Southern Ocean during the austral summer of 2017 and suggest a key role of microdiversity in separating the different water masses. Further metagenomic analysis, as well as a temporal re-sampling of the same area (seasonally or annually), will better clarify the specific functions of the dominant taxa living there.

## Figures and Tables

**Figure 1 microorganisms-11-00702-f001:**
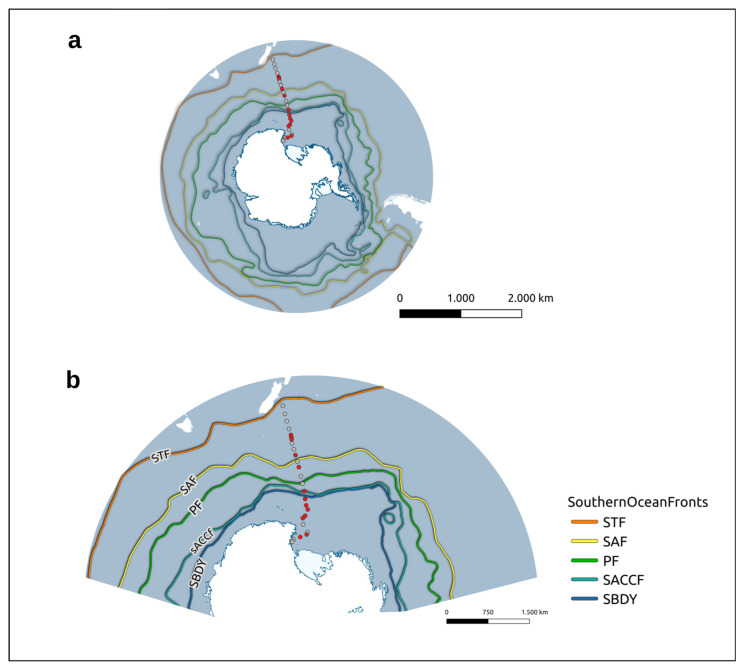
Transect between New Zealand and the Ross Sea sampled during the oceanographic campaign: (**a**) Distribution of the sampled stations (gray dots) and of the stations for which 16S rRNA libraries were obtained (red dots). In the figure are also reported the current fronts crossing the transect: STF—Sub-Tropical front; SAF—Sub-Antarctic Front; PF—Polar Front; SACCF—Southern Antarctic Circumpolar Current front; SBDY—Southern Boundary of the ACC; (**b**) Zoomed view of the stations covered within the transect with their name.

**Figure 2 microorganisms-11-00702-f002:**
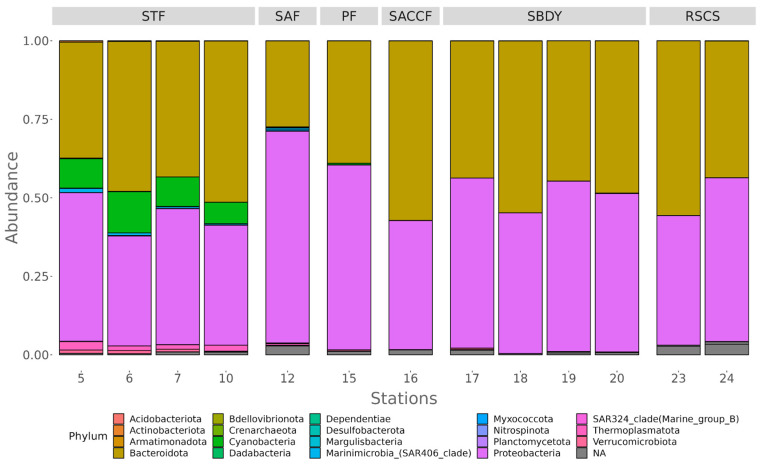
Diversity at the phylum level for all the stations considered. On top of the figure are the fronts to which each sampling station can be assigned. STF—South Tropical Front; SAF—Sub-Antarctic Front; PF—Polar Front; SACCF—Southern ACC Front; SBDY—Southern Boundary Front; RSCS—Ross Sea Continental Shelf.

**Figure 3 microorganisms-11-00702-f003:**
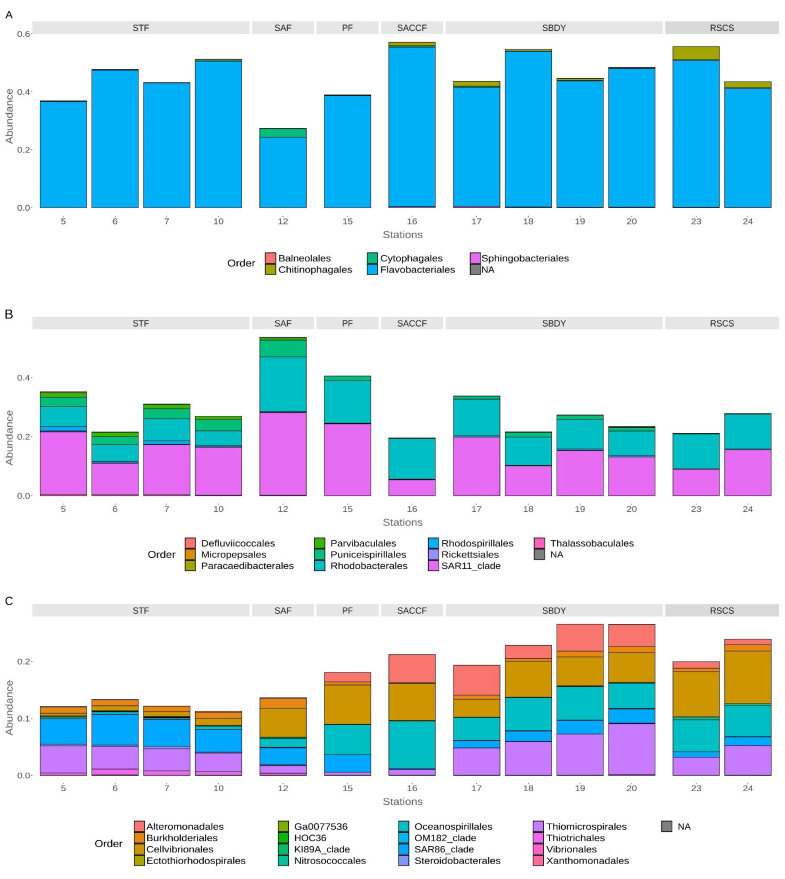
Order level distribution of the 16S rRNA diversity within the classes Bacteroidia (**A**), Alphaproteobacteria (**B**) and Gammaproteobacteria (**C**) for the stations.

**Figure 4 microorganisms-11-00702-f004:**
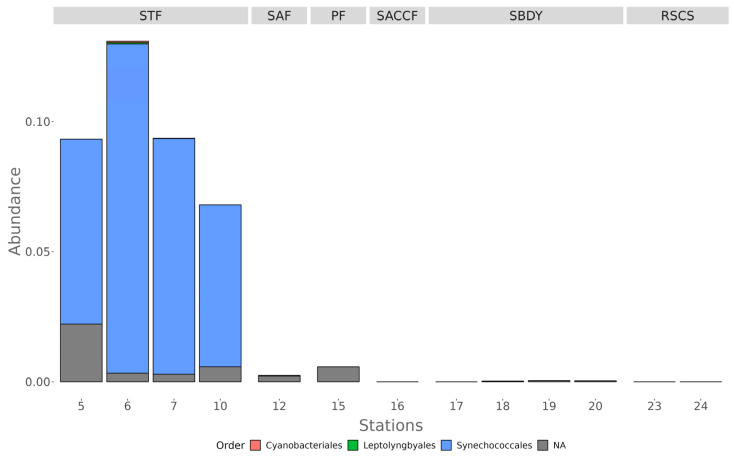
Order level distribution of the 16S rRNA diversity within the phylum Cyanobacteria.

**Figure 5 microorganisms-11-00702-f005:**
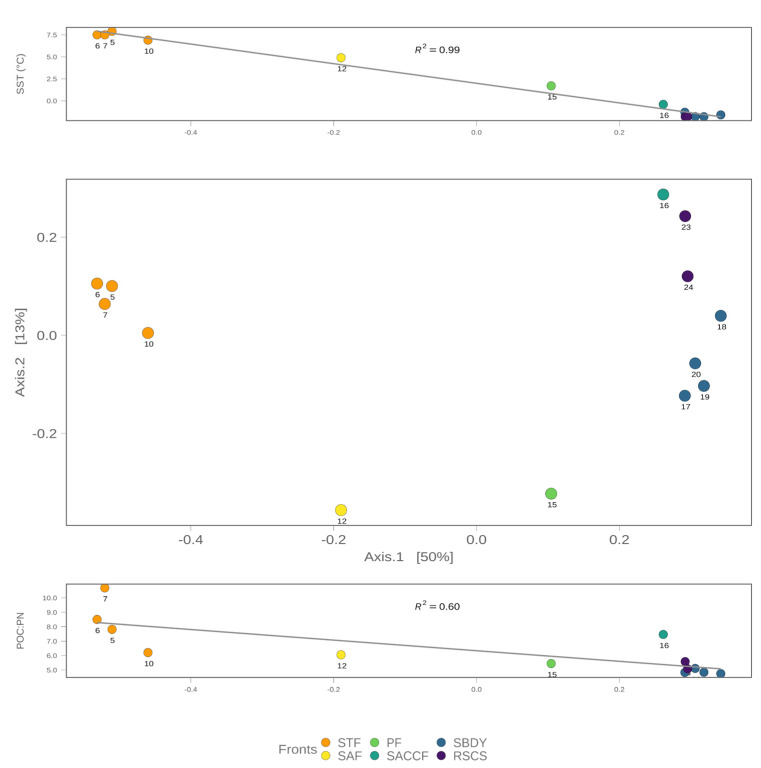
Principal Coordinate Analysis using a weighted Jaccard similarity distance illustrating the shifts in the community structure according to Southern Ocean waterfronts. SST—Sea Surface Temperature; POC:PN—Ratio between Particulate Organic Carbon and Particulate Nitrogen; STF—Sub-Tropical Front; SAF: Sub-Antarctic Front; PF—Polar Front; SACCF—Southern Antarctic Circumpolar Current Front; SBDY—Southern Boundary Front; RSCS—Ross Sea Continental Shelf. The amount of variance explained by each axis is reported within square brackets.

**Figure 6 microorganisms-11-00702-f006:**
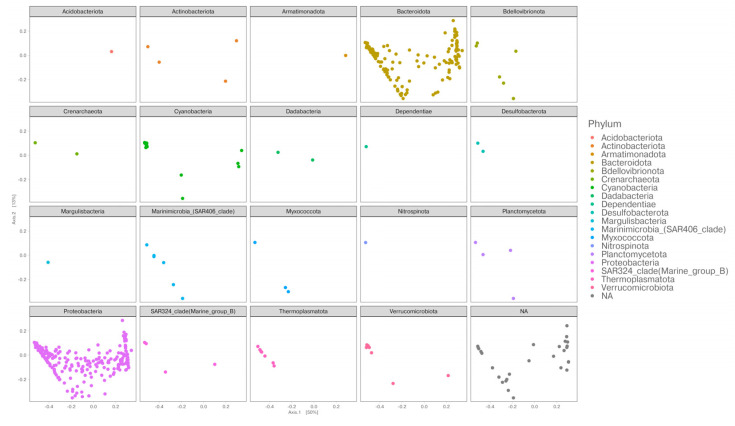
Phylum level load of the ASVs that mainly contribute to the distribution of the sampled stations within the PCoA ordination.

**Figure 7 microorganisms-11-00702-f007:**
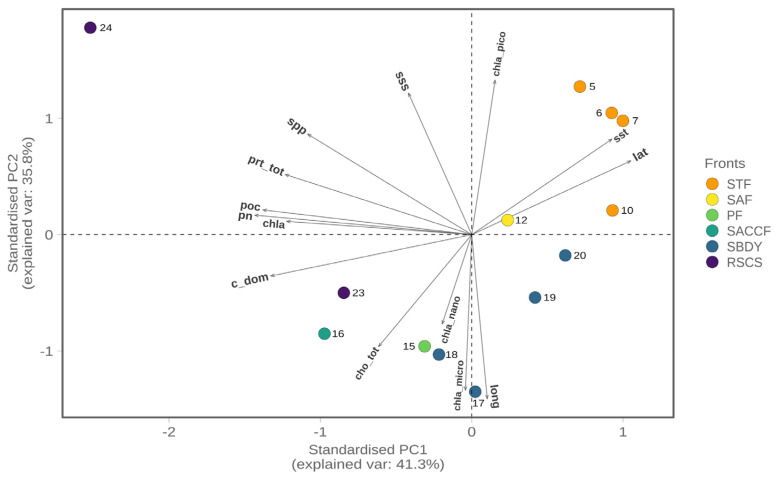
Principal Component Analysis biplot with each individual sample colored by front. STF—Sub=Tropical Front; SAF—Sub-Antarctic Front; PF—Polar Front; SACCF—Southern Antarctic Circumpolar Current Front; SBDY—Southern Boundary Front; RSCS—Ross Sea Continental Shelf. Lat—latitude; sst—sea surface temperature; sss—sea surface salinity; spp—surface primary productivity; prt_tot—total proteins; poc—particulate organic carbon; cho_tot—total carbohydrates; pn—particulate nitrogen; c-dom—carbon fraction of dissolved organic matter; chl-a—chlorophyll-a; chla-micro—microfraction of chlorophyll-a; chla-nano—nanofraction of chlorophyll-a; chla-pico—picofraction of chlorophyll-a.

**Table 1 microorganisms-11-00702-t001:** Coordinates and environmental parameters of sequenced stations. SST—Sea Surface Temperature; SSTA—Sea Surface Temperature Anomaly; SSS—Sea Surface Salinity; CS—Current Speed; WS—Wind Speed; NSAF—North Sub-Antarctic Front; SSAF—South Sub-Antarctic Front; PF—Polar Front; SACCF—Southern Antarctic Circumpolar Current Front; RSCS—Ross Sea Continental shelf; STF—Sub-Tropical Front; SAF—Sub-Antarctic Front; sBdy—Southern Boundary Front.

Station	Latitude [°N]	Longitude [°E]	SST [°C]	SSTA [°C]	SSS [PSU]	cs [m/s]	ws [km/h]	Antarctic Current	Oceanic Front
5	−51.8092	175.0061	7.9	−0.8	34.4	0.32	62	NSAF	STF
6	−52.3100	175.0749	7.5	−0.7	34.3	0.29	14	NSAF	STF
7	−52.7343	175.0708	7.5	−0.7	34.3	0.24	15	NSAF	STF
10	−56.1723	175.8037	6.9	0	33.9	0.1	26	SSAF	STF
12	−58.8349	176.6923	4.9	−0.3	34	0.06	37	SSAF	SAF
15	−64.2124	178.3410	1.7	0.6	33.8	0.24	39	PF	PF
16	−65.9702	178.5408	−0.4	0	33.6	0.01	30	SACCF	SACCF
17	−67.4229	178.9109	−1.3	−0.3	33	0.01	13	SACCF	sBdy
18	−68.3780	179.7789	−1.6	−0.5	33.2	0.02	16	SACCF	sBdy
19	−69.7571	177.9027	−1.8	−0.6	33.8	0.03	17	SACCF	sBdy
20	−70.2680	176.2240	−1.8	−0.7	34.3	0.03	7	SACCF	sBdy
23	−74.0641	178.1007	−1.8	−1.5	34.5	0.05	17	RSCS	RSCS
24	−74.4742	172.4104	−1.8	−1.7	35.2	0.03	23	RSCS	RSCS

**Table 2 microorganisms-11-00702-t002:** Biogeochemical parameters determined for the sequenced stations. spp—surface primary productivity; prt_tot—total proteins; cho_tot—total carbohydrates; PN—Particulate Nitrogen; POC—Particulate Organic Carbon; C-DOM—C fraction of Dissolved Organic Matter; Chl-a—Chlorophyll-a; Micro-chla—Microfraction of chlorophyll-a; Nano-chla—Nanofraction of chlorophyll-a; Pico-chla—Picofraction of chlorophyll-a.

Station	ssp [mg C/m^2^ Day]	prt_tot [µg/L]	cho_tot [µg/L]	PN [µg/L]	POC [µg/L]	C-DOM [µg/L]	Chl-a [µg/L]	Micro- Chla [%]	Nano- Chla [%]	Pico-Chla [%]
5	398.14	66.84	41.60	9.00	69.77	11.11	0.32	12.98	30.81	56.20
6	323.15	56.58	35.98	6.71	57.22	8.12	0.41	1.63	54.14	44.23
7	290.04	48.64	33.71	4.91	52.76	8.70	0.58	12.66	40.06	47.28
10	282.89	46.76	18.05	7.38	46.41	8.49	0.69	14.22	64.64	21.14
12	265.15	56.56	34.37	14.87	90.08	11.72	1.44	15.28	56.38	28.33
15	211.15	56.81	58.67	17.94	97.71	16.74	1.56	23.13	62.66	14.21
16	275.82	98.02	63.21	28.90	216.08	17.41	1.06	24.51	56.68	18.81
17	285.45	49.86	59.06	16.59	79.81	14.90	0.55	31.24	51.44	17.32
18	249.01	53.63	54.32	22.27	106.55	12.12	1.44	21.42	55.29	23.29
19	244.42	36.57	43.70	10.49	50.87	10.08	0.63	19.80	51.32	28.88
20	103.30	23.87	31.64	8.92	45.32	10.25	0.55	14.85	54.89	30.26
23	475.60	71.14	64.82	20.80	116.38	18.08	0.99	20.91	53.31	25.78
24	943.99	124.22	34.73	54.01	274.66	19.13	2.47	3.15	49.46	47.39

## Data Availability

The sequences analyzed in the present study are publicly available through the European Nucleotide Archive (ENA) with bioproject accession number PRJEB45048. A complete R script describing the analysis can be found in the GitHub repository https://github.com/giovannellilab/Cordone_et_al_Southern_Ocean_microbial_diversity, accessed on 30 January 2023 and released with Zenodo with https://doi.org/10.5281/zenodo.7584581, accessed on 30 January 2023.

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
