# Peer review of "Surface Bacterioplankton Community Structure Crossing the Antarctic Circumpolar Current Fronts"

_microorganisms, 2023, doi:10.3390/microorganisms11030702_

Round 1

Reviewer 1 Report

Dear authors:

The manuscript is very well written and here are only some suggestions to finish the text.

Line 30 – include in the abstract the year of the data sampling (2017). There are two dots at the phrase end.

Keywords – delete “Antarctic Circumpolar Fronts” since it is present on the title.

Line 197 – I could not access the site https://github.com/giovannel-

lilab/Cordone_et_al_Southern_Ocean_microbial_diversity. May be because of internet quality here in Antarctica, but please check.

Figure 1 could be larger, being an option to put the sections A and B one above the other. In all the Tables it is important to show the abbreviation in a legend below tables.

Figure 3 – in the legend it is important to explain what is A, B and C and in Lines 237 and 248 refer to them as (Figure 3A) and (Figure 3B).

Figure 4 – Explain abbreviations.

Figure 5 – Explain abbreviations.

Line 288 – change “...is among one of the...” to “ … is the… ”.

Line 358 – change “Taxa” to “taxa”.

Line 371 – include “(Italy)” after the “Ministero dell’Istruzione, dell’Università e della Ricerca”.

Round 2
